# Feasibility of Proton Beam Therapy as a Rescue Therapy in Heavily Pre-Treated Retinoblastoma Eyes

**DOI:** 10.3390/cancers13081862

**Published:** 2021-04-13

**Authors:** Eva Biewald, Tobias Kiefer, Dirk Geismar, Sabrina Schlüter, Anke Manthey, Henrike Westekemper, Jörg Wulff, Beate Timmermann, Petra Ketteler, Stefan Schönberger, Klaus A. Metz, Saskia Ting, Sophia Göricke, Nikolaos E. Bechrakis, Norbert Bornfeld

**Affiliations:** 1Department of Ophthalmology, University Hospital Essen, University Duisburg Essen, 45122 Essen, Germany; tobias.kiefer@uk-essen.de (T.K.); sabrina.schlueter@uk-essen.de (S.S.); anke.manthey@uk-essen.de (A.M.); henrike.westekemper@uk-essen.de (H.W.); nikolaos.bechrakis@uk-essen.de (N.E.B.); norbert.bornfeld@uk-essen.de (N.B.); 2Department of Particle Therapy, West German Proton Therapy Centre Essen (WPE), University Hospital Essen, German Cancer Consortium (DKTK), 45122 Essen, Germany; dirk.geismar@uk-essen.de (D.G.); joerg.wulff@uk-essen.de (J.W.); beate.timmermann@uk-essen.de (B.T.); 3Department of Pediatric Hematology and Oncology, University Duisburg Essen, 45122 Essen, Germany; petra.ketteler@uk-essen.de (P.K.); stefan.schoenberger@uk-essen.de (S.S.); 4Institute of Pathology, University Hospital Essen, University Duisburg-Essen, 45122 Essen, Germany; klaus.metz@uk-essen.de (K.A.M.); saskia.ting@uk-essen.de (S.T.); 5Department of Diagnostic and Interventional Radiology and Neuroradiology, University Hospital Essen, University Duisburg-Essen, 45122 Essen, Germany; sophia.goericke@uk-essen.de

**Keywords:** EBRT, subsequent primary malignancies, eye preserving therapy, in-field malignancies

## Abstract

**Simple Summary:**

A variety of therapies are available for the treatment of retinoblastomas. Nevertheless, despite exhaustion of all therapeutic methods, refractory or recurrent courses of the disease occur. In eyes with a function worthy of preservation radiation therapy may become unavoidable. Proton beam therapy, compared to conventional photon-based radiotherapy, is a highly conformal form of radiation therapy with a high biological effectiveness with a simultaneously reduced probability of radiation-related side-effects and induction of secondary primary malignancies. The aim of our retrospective study was to evaluate the efficacy of proton beam therapy as rescue therapy in 15 heavily pretreated retinoblastoma eyes. In our retrospective series of a highly negatively selected patient population, we were able to preserve 60% of the eyes with a manageable side effect profile. A cataract, as the most common long-term complication, was evident in 44.4% of the preserved eyes. There was no in-field second tumor manifestation during follow-up, therefore the preliminary data of this study and series published by others suggest that the risk is significantly lower after proton beam therapy compared to conventional external beam radiation therapy using photons.

**Abstract:**

Despite the increased risk of subsequent primary tumors (SPTs) external beam radiation (EBRT) may be the only therapeutic option to preserve a retinoblastoma eye. Due to their physical properties, proton beam therapy (PBT) offers the possibility to use the effectiveness of EBRT in tumor treatment and to decisively reduce the treatment-related morbidity. We report our experiences of PBT as rescue therapy in a retrospectively studied cohort of 15 advanced retinoblastoma eyes as final option for eye-preserving therapy. The average age at the initiation of PBT was 35 (14–97) months, mean follow-up was 22 (2–46) months. Prior to PBT, all eyes were treated with systemic chemotherapy and a mean number of 7.1 additional treatments. Indication for PBT was non-feasibility of intra-arterial chemotherapy (IAC) in 10 eyes, tumor recurrence after IAC in another 3 eyes and diffuse infiltrating retinoblastoma in 2 eyes. Six eyes (40%) were enucleated after a mean time interval of 4.8 (1–8) months. Cataract formation was the most common complication affecting 44.4% of the preserved eyes, yet 77.8% achieved a visual acuity of >20/200. Two of the 15 children treated developed metastatic disease during follow-up, resulting in a 13.3% metastasis rate. PBT is a useful treatment modality as a rescue therapy in retinoblastoma eyes with an eye-preserving rate of 60%. As patients are at lifetime risk of SPTs consistent monitoring is mandatory.

## 1. Introduction

Hilgartner was assigned the first proof of efficacy of radiotherapy for the treatment of retinoblastoma in 1903 [1]. Since then, external beam radiotherapy (EBRT) has become an essential element in the armamentarium for eye-preserving therapy of retinoblastomas [2] until long-term studies showed that EBRT increases the already enhanced risk for subsequent primary tumors (SPT) in hereditary retinoblastoma survivors significantly [3,4,5,6,7].

In search of an alternative to EBRT [8,9] systemic chemotherapy was introduced in the treatment of retinoblastoma in the 1990s. [10]. Systemic chemotherapy in the application as chemoreduction with additional local procedures such as brachytherapy, cryotherapy or laser therapy achieved satisfactory tumor control avoiding the adverse events of EBRT [11]. Recent developments in eye salvaging treatment of retinoblastoma include local routes of chemotherapy (intra-arterial and intravitreal), which have led to a reduced use of systemic chemotherapy [12].

Despite this wide range in eye salvaging treatment of retinoblastoma, risks remain that all therapeutic options are exhausted and the tumor is insufficiently controlled. This is particularly challenging if the patient has only one remaining functional eye. In these cases, radiation therapy may be the last option to salvage the affected eye knowing the high radiation sensitivity of retinoblastomas [13]. In this case, it is important to keep the radiation field as small as possible to reduce the risk of radiation induced SPTs. Proton beam therapy (PBT) is a well-studied treatment modality in other pediatric and ocular tumors such as pediatric brain tumors or uveal melanoma [14,15]. PBT has unique physical properties compared to ERBT using photons. The lack of an exit dose (Bragg peak) and a small penumbra results in less radiation damage to collateral tissue [16]. This possibly reduces not only ocular side effects such as cataract development, orbital fat atrophy, and the incidence of bone growth abnormalities, but hopefully also the incidence of subsequent primary malignancies as the size of target volume in PBT is reduced compared to EBRT. So far, only few papers have been published on the effectiveness of proton beam therapy in hereditary retinoblastoma [16,17,18,19,20,21,22]. We report our experiences with PBT as a rescue therapy in advanced retinoblastoma disease as a last option for eye-preserving therapy.

## 2. Materials and Methods

We identified all patients treated with PBT at our institution from February 2016 to August 2018 with at least 12 months of follow-up information, provided that the eyes could be preserved up to the required minimum follow-up All included patients were pretreated retinoblastoma eyes, with PBT being a rescue therapy for failure or lack of feasibility of all other established therapy modalities. In particular, PBT was the only therapeutic alternative to enucleation in the majority of cases. Patients who showed an indication for orbital radiation via PBT due to histopathological risk factors after enucleation were excluded (*n* = 6). Patient data were reviewed for age at the beginning of PBT, gender, laterality, hereditary, follow-up time, visual acuity if available and initial grouping of the International Retinoblastoma Classification (IRCB), if applicable. Concerning the therapy course, we recorded number and type of pretreatments, incidence and type of therapy complications, tumor control rate, tumor recurrences, globe saving rates and metastatic disease. In accordance with the regulatory requirements that apply to our institution, the Internal Review Board (IRB) approval was not required for this retrospective study.

### Treatment Technique

Before PBT was planned, all patients underwent an extensive ophthalmological examination under anesthesia in order to record and document the exact extent of the tumor(s) intended to treat. Depending on clinical findings, staging examinations were performed. The indication for PBT was determined in an interdisciplinary tumor board. Two tantalum markers were placed on the nasal sclera to ensure the correct positioning of the affected eye during PBT. For treatment patients were immobilized with a base of skull headframe (BOS™) and a thermoplastic mask in supine position. Computed tomography (CT) and magnetic resonance imaging (MRI) in treatment position under deep anesthesia were performed to plan the proton treatment. In the first cases of PBT with the obligation to provide proof of effectiveness only the entire eye could be treated. With advanced planning security and refinement of the technique, we achieved an optimized target volume in order to treat the tumors with a precise safety margin. The exact target volume was defined in cooperation between the treating ophthalmologist and the radiation oncologist, with the attempt to keep the treated volume as small as possible to minimize the risk of SPT in the radiation field. In the beginning the average safety margin around the tumor (gross tumor volume = GTV) intended to treat was 5 mm for clinical target volume (CTV) and 3 mm for planning target volume (PTV). The CTV was anatomical modified to the eye globe. For large tumors, in an anteriorly location or with vitreous seeding the whole retina and vitreous body were targeted for PBT. Whenever possible, the lens was spared from the irradiation in order to prevent premature cataract development with complicated tumor control. The treatment planning system (TPS) RayStation (version 6/7, RaySearch laboratories, Stockholm, Sweden) was used [23]. The treatment was applied by the ProteusPlus therapy machine (IBA, Lovaine-La Nueve, Belgium) operated in the spot-by-spot pencil beam scanning mode (spot size: 2.6–5.7 mm in air (beam sigma)) or in the uniform scanning mode. Apertures were used for uniform scanning and in some cases for pencil beam scanning. The irradiation itself was performed under general anesthesia. Conventionally fractionated irradiation by typically one lateral field was performed with a dose of 2 Gy per day for 25 days of irradiation, corresponding to a total dose of 50 Gy.

## 3. Results

We analyzed 15 eyes of 13 bilaterally affected patients, 10 were male and 3 were female. The mean age at initiation of PBT was 35 months (14–97 months), standard deviation was 28.2 months All but two eyes were the only functional eyes. In another two patients both eyes were treated by PBT. Three children had familial retinoblastoma with a proven RB1 mutation (23%). The mean follow-up time was 22 months (2–46 months), standard deviation was 15.0 months At initial presentation, patients were classified using the International Classification System for Retinoblastoma (ICRB) [24] as follows: ICRB A *n* = 3 (20%), ICRB B *n* = 5 (33.3%), ICRB D *n* = 3 (20%), ICRB E *n* = 1 (6.7%) and unknown in pretreated eyes *n* = 3 (20%). In all patients, systemic intravenous chemotherapy was the primary therapy. In nine eyes additional transpupillary thermochemotherapy was performed depending on tumor location and volume. Another two eyes with vitreous seeding received intravitreal chemotherapy (3–10 procedures per eye) in addition to systemic chemotherapy. The mean number of additional local or systemic treatments after failure of intravenous chemotherapy was 7.1 per eye (0–18). Prior PBT, 10 eyes received intra-arterial chemotherapy (1–5 procedures per eye), 7 eyes had intravitreal chemotherapy (1–6 procedures per eye) and 5 eyes were treated with ruthenium- plaque brachytherapy. Table 1 gives an overview of these baseline data.

Indication for PBT was recurrence or progression after intra-arterial chemotherapy in 3 eyes and a lack of technical feasibility of repeated intra-arterial chemotherapy in 10 eyes. In these cases, either a technical failure of repeated ophthalmic artery catheterization or a retrograde blood flow with insufficient blush of the affected eye was revealed. Two eyes presented with a diffuse infiltrating anterior retinoblastoma with massive involvement of the anterior segment.

PBT was performed as described above. Figure 1 displays the irradiation plan of a tumor recurrence at the posterior pole with the corresponding MRI scan and RETCAM™ images before and after irradiation.

### 3.1. Rate of Eye Preservation and Tumor Control

Six eyes (40%) could not be preserved after PBT and had to be enucleated after a mean time interval of 4.8 months (2–8 months). Indications for secondary enucleation were tumor recurrence at the optic nerve head with the risk of optic nerve involvement or choroidal involvement in three eyes, persisting exudative retinal detachment and intraocular bleeding with loss of tumor control and light perception in two eyes and a ciliary body insufficiency and globe hypotony after irradiation of the complete anterior segment with again loss of light perception and tumor control in another eye. Three of those six eyes were enucleated at our institution. Within the histopathological specimens viable tumor cells were found in two eyes, as it was suspected on clinical examination. Three eyes were enucleated elsewhere. In one of those eyes, no viable tumor cells were detected whereas the second case showed viable tumor cells with a massive choroidal infiltration. In the third case the course was unfavourable, since the parents refused enucleation and were treated with further intraarterial chemotherapy elsewhere in the attempt to preserve the eye. By the time the eye was finally enucleated the child had already developed systemic metastases as a result of optic nerve and choroidal infiltration and eventually died from metastatic disease. Figure 2 shows the RETCAM™ images, macroscopic images, MRI images and the histopathological slides of one patient with proven viable tumor cells after enucleation.

Two eyes demanded further therapy after PBT for tumor recurrence. One eye had to be treated with another four injections of intravitreal chemotherapy and a plaque brachytherapy. The other eye received three more injections of intravitreal chemotherapy after PBT was completed. Recurrence therapy in these cases were initiated 3 and 6 months after PBT, respectively. Table 2 summarizes the long-term follow-up with radiogenic complications, additional treatment after PBT, globe salvage rate, long-term survival and histopathological workup, if applicable.

### 3.2. Radiation Induced Side Effects

Radiation induced cataract occurred in four of the nine preserved eyes, where the lens could not be spared due to tumor localization. The mean time interval between PBT and occurrence of cataract was 22.5 months (16–25 months). In three eyes, uncomplicated lensectomy was performed after a mean time of 31.7 months (30–33 months). Concerning other serious complications following PBT, one eye with involvement of the anterior segment developed a chronic ocular surface disease, which ultimately lead to a perforated corneal ulcer with the need of an emergency penetrating keratoplasty 19 months after PBT. Figure 3 shows the initial findings with a diffuse infiltrating retinoblastoma and the further clinical course with complete tumor regression and emergency keratoplasty during follow-up. So far, disease remained stable, however due to chronic surface problems VA did not improve to more than 20/200.

We observed no in-field secondary malignancies during follow-up.

### 3.3. Visual Outcome

In nine eyes visual acuity (VA) during follow-up could be recorded. Seven of these eyes had a best-corrected VA ≥ 20/200 ranging from 20/100 to 20/25. Two eyes had a VA of less than 20/200.

### 3.4. Metastatic Disease

Two of the 15 children treated developed metastatic disease during follow-up, resulting in a 13.3% metastasis rate. In one patient, the parents’ refusal to perform the necessary enucleation led to the development of systemic disease. This patient has already been explained in detail above. The second patient with a diffuse infiltrating anterior retinoblastoma, developed paravertebral, intraspinal and mandibular metastatic disease 7 months after completion of PBT. Treatment with high-dose chemotherapy, autonomous stem cell transplantation and complementary radiation therapy resulted in restitutio ad integrum. So, up to now the child is alive and recurrence free.

## 4. Discussion

Without ignoring the well-known serious long-term complications of EBRT in hereditary retinoblastoma, external beam radiotherapy may be the only option to preserve a therapy-refractory retinoblastoma. This is especially true in cases where the only seeing eye is affected. Most common histological types of SPTs in the irradiation fields are osteosarcoma, rhabdomyosarcoma, leiomyosarcoma, other soft tissue sarcomas and meningeoma. In order to minimize the risk of radiogenic induction of SPTs, radiation using a linear accelerator and a lateral D-shaped field (Schipper technique [25]) has been largely abandoned and all efforts have been concentrated on achieving conformal irradiation with the smallest possible planning target volume (PTV). Technical possibilities to achieve this goal could be conformal irradiation with photons or proton beam therapy, whereby the latter technique is most likely to achieve these targets [2,26].

The average risk to develop a SPT after radiation therapy in hereditary retinoblastoma patients is approximately 1% per year [27]. As the majority of SPTs occur in the radiation field the physical properties of protons compared to photons with a steep dose fall-off at the end of the irradiation path and a minimal penumbra may significantly reduce the risk of radiogenic induced tumors. With PBT, a major part of the bony orbit can be spared from the planning target volume in PBT. In 2014, Sethi et al. published their results on 86 retinoblastoma patients, either treated with photon or proton beam therapy [19]. They found the 10-year cumulative incidence of radiation-induced or in-field second malignancies to be significantly different between radiation modalities, being 0% in the PBT group vs. 14% in the EBRT group (*p* = 0.015) for in-field second malignancies. Therefore, they concluded that PBT significantly lowers the risk of SPTs compared to EBRT [19]. In 2005, Krengli et al. compared different tumor localizations regarding the optimization of proton radiation therapy and correlated the isodose distribution of affected and unaffected structures [22]. In all cases of posterior–central, nasal and temporal tumor locations, they achieved homogeneous target coverage with true lens sparing and reduction of the radiation dose to the bony orbit. They concluded that PBT minimizes the risk of SPTs and the risk of cosmetic and functional impairments [22]. It is widely accepted that the only way to increase survival in patients with SPT is prompt and complete resection of the tumors, therefore early diagnosis is essential [27,28]. The bony lateral orbital wall, which inevitably is included in the planned target volume in PBT, is very easily accessible for clinical controls, so that the chances for an early diagnosis are high and an adequate therapy can be initiated promptly.

Conventional EBRT of hereditary retinoblastoma with a linear accelerator using the Schipper technique results in severe midface growth inhibition with corresponding cosmetic consequences. Mourits et al. could show in 195 retinoblastoma survivors that EBRT resulted in worse cosmetic outcomes compared to enucleation alone [29]. They demonstrated that patients treated with EBRT had significantly more superior sulcus volume deficiencies than those treated with enucleation alone [29]. In addition, Mouw et al. [18] compared the MRI-based orbital heights and widths of retinoblastoma patients treated with PBT and/or enucleation in a cohort of 12 participants. In their small series the comparison of the group with unilateral enucleation and PBT on the contralateral side revealed that the orbital height and/or width was >1 mm smaller on the enucleated side. Thus, there is evidence that PBT, when compared to EBRT or enucleation, might be superior in terms of cosmetic results potentially attenuating facial asymmetries such as midfacial hypoplasia. In a subset of patients with larger tumors in whom the posterior chamber of the eye and the median orbital wall were included in the target volume, however, growth restriction of the orbit had then occurred in this series.

The efficacy of PBT in the treatment of naive and previously treated retinoblastoma eyes has been investigated in several studies. Jung et al., 2018 published their results on proton beam therapy in four retinoblastoma eyes with vitreous seeds [16]. All four eyes were group D eyes with clouds. Tumor and vitreous seed regression could be achieved in 50% of these cases. The remaining two eyes could not be preserved. Follow-up of the two salvaged eyes was 31 months on average with no radiation associated side effects or SPTs in the radiation field. A larger series with long-term results on PBT in 60 eyes of 49 retinoblastoma patients was published in 2014 by Mouw et al. [20]. In this series approximately half of the patients had previously been treated with systemic chemotherapy. After a mean follow-up of 8 years no patient had died of metastatic disease or developed an in-field SPT. In the long-term course 82% of the eyes could be preserved with a higher enucleation rate in group C or D eyesmainly due to tumor progression and rarely because of radiation complications. The main complication was cataract progression as the tumor progression rate was low and occurred within in the first two years after PBT. The average additional follow-up was 12 years, with no long-term effects on visual acuity or hormonal dysfunction observed in the cohort. A closer look at the primary with PBT treated group A and B eyes revealed that the outcome in those subgroups was very favorable. Eye preservation rate was very high, as almost 90% of the treated tumors could be permanently controlled with PBT. They concluded that with reference to radiation toxicity and the favorable outcomes PBT might be appropriate as first-line treatment for treatment naïve group A and B eyes. Referring to the results of this study and considering the increasingly better technical possibilities to safely irradiate a small PTV, PBT may be considered as first line therapy in cases with tumors close to the macula or the optic disc.

In our cohort, we could maintain 60% of the eyes treated with PBT over a mean follow-up period of 22 months (2–46 months). 40% of the analyzed eyes could not be preserved during the follow-up period. The main reasons for enucleation were tumor progression in four eyes, vitreous hemorrhage subsequent to radiation retinopathy and ocular hypotony with a loss of light perception in another two eyes. As reported in previous studies those were heavily pretreated eyes with advanced tumor stages and almost globe-filling relapses, PBT being the last treatment option prior enucleation. This patient cohort unfortunately did not benefit from PBT, therefore the indication for PBT should be very strict as the chances of success in eyes with a massive pretreated relapse appear to be rather low. Unfortunately, one of those children died during follow-up as the parents refused secondary enucleation strongly indicated by tumor progression and optic nerve involvement. On the other hand, the 60% of the patients in our cohort, where we could maintain the eye, showed a very good tumor control rate and significantly fewer side effects compared to previously used EBRT. Those were eyes with a more circumscribed tumor recurrence and therefore treated with a very small radiation field after refinement of the technique.

In our series, a decrease in VA due to radiation-induced cataract was observed in 26.7% of all treated eyes (4/15) after 21 months on average. In all these patients, the lens or the anterior segment could not be spared due to tumor localization. However, three of these four eyes had a complication-free lensectomy. In one patient with a diffuse infiltrating anterior retinoblastoma, the anterior segment had to be included in the PTV resulting in corneal ulceration with perforation that was treated successfully with penetrating keratoplasty. The transplant is stable since more than 17 months now, however the patient suffers from recurrent corneal erosion.

In general, the development of a radiation-induced cataract is a minor problem in the long-term course of irradiated retinoblastoma eyes, as surgery can be performed without major complications in the majority of cases. In 2017, Kim et al. published their results on cataract surgery in five eyes of five patients after treatment for retinoblastoma without any intra- or postoperative complications and no signs of tumor recurrence [30]. A larger series with pars plana lensectomy and intraocular lens implantation in pediatric radiation-induced cataracts in retinoblastoma in 16 eyes of 12 children was published by Miller et al. in 2005 [31]. In this series, no viable tumor cells were found in the vitrectomy specimen with a mean latency of 42 months after termination of EBRT. Likewise, they observed no late recurrence, orbital tumor or metastatic disease during follow-up. A gain in VA was documented in 69% while main complications were transient macular edema in 31% and iridocyclitis in 19% of all treated eyes.

The advantages and disadvantages of PBT must be distinguished against the results of systemic and local chemotherapy [26]. Systemic chemotherapy in advanced tumor stages is often associated with a higher recurrence rate, especially if no adjuvant local therapy strategy is used [32]. Furthermore, serious complications and side effects as cytopenia, fever, neutropenia, infection, gastrointestinal symptoms, dehydration and neurotoxicity must be taken into account when planning therapy [33,34]. Regarding new local targeted chemotherapy strategies, mainly intra-arterial (IAC) or intravitreal chemotherapy (IVitC), toxicity aspects and the potential risk of IAC regarding repeated X-ray scans for hereditary disease carriers should not be forgotten [35]. Clinical and preclinical studies of IAC and IVitC could prove a decrease in electroretinogram response, which is indicative of retinal toxicity [36]. Concerning IAC Ravindran et al. just recently published a review article and a meta-analysis of 20 studies with a special focus on metastatic disease and globe salvage [37]. They analyzed more than 1400 eyes of 873 patients. Metastatic disease occurred in 1.6% of the cases and 11.8% of the eyes had to be enucleated. These were, in particular, advanced retinoblastoma eyes, where the globe salvage rate was 35.6% only. Concerning local complications retinal and choroidal ischemia were reported in 13.1%, retinal detachment in 23.3%, chorioretinal atrophy in 6.4% and vitreous and vitreous hemorrhage in 11.9%. All those serious side effects were associated with a severe decrease in visual acuity. Thus, IAC is not without complications and the long-term consequences of repeated scanning during IAC, especially in hereditary retinoblastoma patients, are not yet foreseeable.

## 5. Conclusions

In conclusion, PBT is an effective rescue treatment in advanced hereditary retinoblastoma with no other treatment option in eyes with a function worthy of preservation. Preliminary data of this study and series published by others suggest that the risk of SPTs was significantly lower and the growth inhibition of the bony orbit was less after PBT compared to conventional EBRT using photons. The results of this study suggest further investigation of the potential use of PBT as a first-line therapy in highly selected cases.

## Figures and Tables

**Figure 1 cancers-13-01862-f001:**
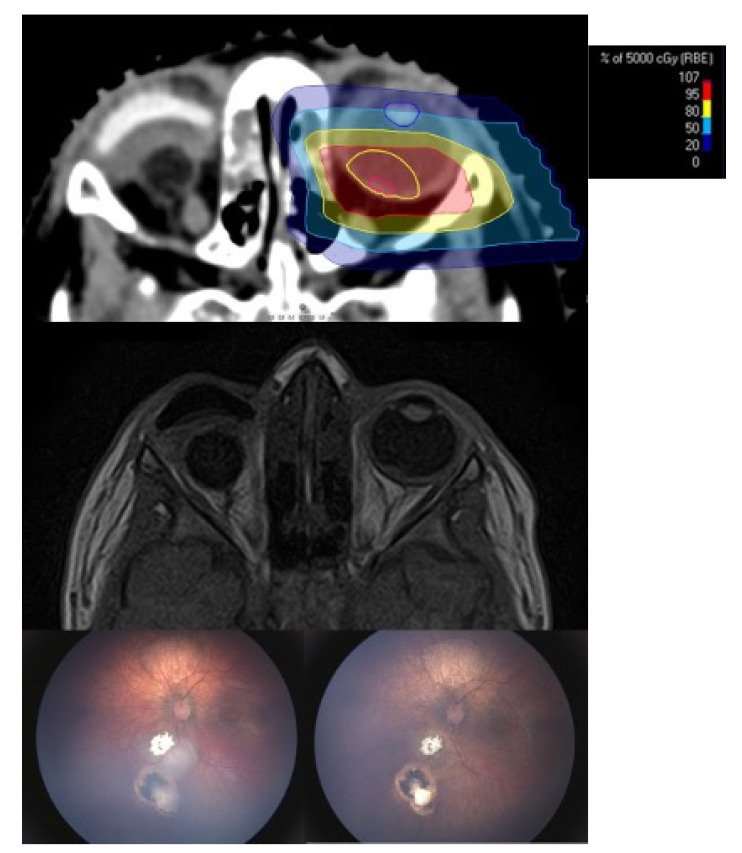
(Case 2): 21 old month male patient before and after proton beam therapy. The top figure shows the irradiation plan (gross tumor volume (GTV): red and clinical target volume (CTV): yellow) and middle figure the corresponding magnetic resonance imaging (MRI) before proton beam therapy (PBT). Note the sparing of the lens and the small bony window. The lower left picture shows two recurrent tumors close to the optic disc. Intraarterial chemotherapy was not feasible due to insufficient visualization of the ophthalmic artery. The lower right picture shows the result after a follow-up of 16 months. Best-corrected visual acuity was 20/40 Snellen acuity.

**Figure 2 cancers-13-01862-f002:**
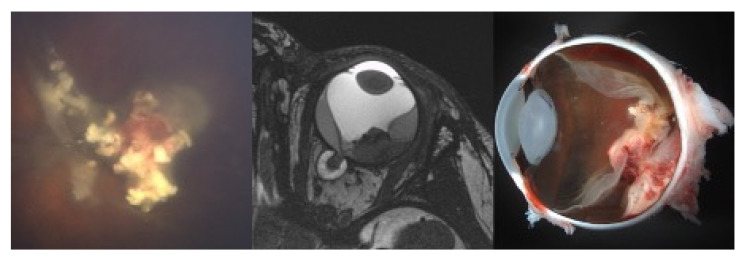
(Case 11): The first picture shows a clinical evident tumor recurrence of the centrally located type III regression 8 months after proton beam therapy. The second picture shows the corresponding MRI images with the prominent tumor and a total exudative retinal detachment. The third picture shows the macroscopic images after enucleation with a visible massive choroidal infiltration only detected by histopathological workup of the specimen.

**Figure 3 cancers-13-01862-f003:**
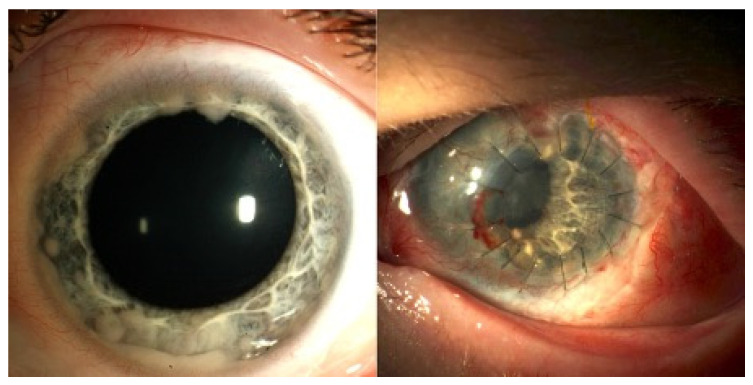
(Case 7): The first picture shows the anterior segment involving tumor recurrence. This was a late recurrence with the initiation of proton therapy at 97 months of age. The tumors showed total regression. However, the child suffers from chronic ocular surface disease due to anterior segment radiation. Emergency keratoplasty had to be performed 19 months after proton beam therapy due to a perforated corneal ulcer. Even after transplantation the eye shows recurrent corneal erosions.

**Table 1 cancers-13-01862-t001:** Initial findings, demographic data and treatment prior to proton beam therapy (PBT).

Patient No.	ICRB	Eye Treated	Age at PBT (Months)	Initial Treatment	Additional Treatment
1	unknown	left	56	IVC	IAC, IVitC, Lc, Cc
2	B	left	21	IVC + TCT	IVitC, Lc, Cc
3	unknown	right	29	IVC + IVitC	IAC, Brachy
4	D	left	25	IVC + TCT	IVitC, Lc
5	D	left	18	IVC	none
6	D	left	50	IVC + IVitC + TCT	Brachy, Lc, Cc
7	A	left	97	IVC	IAC, IVitC, Brachy, Lc, Cc
8	unknown	left	91	IVC	IAC
9	A/B	bilateral	19	IVC + TCT	IAC, Lc, Cc
10	B	left	14	IVC + TCT	IAC, IVitC, Brachy, Lc, Cc
11	E	left	26	IVC	IAC, IVitC
12	B/A	bilateral	23	IVC + TCT	IAC, IVitC, Lc, Cc
13	B	right	15	IVC	none

ICRB = International Classification System for Retinoblastoma; IVC = intravenous chemotherapy; TCT = transpupillary thermochemotherapy; IVitC = intravitreal chemotherapy; IAC = intra-arterial chemotherapy; Brachy = Ruthenium-Plaque Brachytherapy; Lc = Laser coagulation; Cc = Cryocoagulation.

**Table 2 cancers-13-01862-t002:** Long-term follow-up with complications, additional treatment after PBT, globe salvage rate, long-term survival and histopathological workup, if applicable.

Patient No.	Follow-Up in Months	Additional Treatment After PBT	Complications	Globe Salvage	Histopathologic Viable Tumor Cells	Metastatic Disease	**Deaths**
1	2 *	-	Ciliary body failure	no	no	unknown	unknown
2	16	-	-	yes	-	-	-
3	4 *	-	-	no	yes	yes	yes
4	6 *	IVitC, Brachy, IAC (ex domo)	-	no	unknown	unknown	unknown
5	25	IVitC	Dry eye disease	no	yes	no	no
6	46	Lensectomy + IOL	Cataract, radiation retinopathy	yes	-	-	-
7	39	pKPL AMT	Cataract, radiation retinopathy	yes	-	yes	no
8	36	-	-	no	no	no	no
9	39	Lensectomy + IOL	Cataract (bilateral)	yes	-	-	-
10	21	-	Cataract, radiation retinopathy	yes	-	-	-
11	8 *	Triamcinolonesub-tenon	Exudative retinal detachment	no	yes	no	no
12	15	-	-	yes	-	-	-
13	12	-	-	yes	-	-	-

* = enucleation unavoidable before completion of the requested 12 months of follow up after termination of PBT; IVitC = intravitreal chemotherapy; IAC = intra-arterial chemotherapy; IOL = intraocular lens; pKPL = penetrating keratoplasty; AMT = amniotic membrane transplant.

## Data Availability

Aggregated data are available on personal request to the authors.

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
