# Peer review of "Feasibility of Proton Beam Therapy as a Rescue Therapy in Heavily Pre-Treated Retinoblastoma Eyes"

_cancers, 2021, doi:10.3390/cancers13081862_

Round 1

Reviewer 1 Report

This is a well written and important paper. Little literature exists on this topic. I think this paper should be published with very minor revisions. I think it could be published without reviews but I do have minor suggestions/recommendations. 

Intro

  • Considering mentioning decrease in bone growth abnormalities and muscle atrophy in intro as well. These fields give such a small amount of RT to bone and soft tissue and though difficult to publish papers with a picture of a patient showing this, it is minor and great for readers to know this and mentioning in intro as well may be helpful.  
  • Please reword "with thus aggravated tumor control"  I think there is a better word than "aggravated"  

Results

   Change "familiar" to familial 

    CTV- did you retract for anatomical boundaries i.e did CTV extend outside the eye? the orbit? If group C eye did you use this 5 mm margin into vitreous ? or just around/along retina? 

Were apertures used?  Please comment on spot size unless you feel too much detail for readership

Discussion

  • mention brain, muscle minimizations.  Most second malignancies in RT field are sarcomas or brain tumors
  • The authors could consider further discussion on the potential downside to delaying radiation, a clearly effective therapy, for years.  This leads to loss of vision, potential for metastatic spread.  This is based on fear of second malignancy from highly different radiation fields and the risk is much lower with small field proton RT.  Additional bony growth abnormalities are also so much lower.  When I see patients referred for proton RT now, they usually have so little vision and have gradually lost it trying treatments that may work for a period of time but then ultimately show progression and with this progression vision loss.  If the authors could add actual #'s to go with the sentence in the discussion "systemic chemotherapy in advanced tumor stages is often associated with a higher recurrence rate, especially if no local adjuvant therapy is used" this would be of interest to some readers. 

Author Response

Thank you very much for your time and effort in revising our manuscript. We appreciate that and have implemented your suggestions for improvement wherever possible. 
Below you will find our revisions and responses to your suggestions:

Intro

Considering mentioning decrease in bone growth abnormalities and muscle atrophy in intro as well. These fields give such a small amount of RT to bone and soft tissue and though difficult to publish papers with a picture of a patient showing this, it is minor and great for readers to know this and mentioning in intro as well may be helpful.

added in the intro (L 74-75)

Please reword "with thus aggravated tumor control"  I think there is a better word than "aggravated"  

changed to “complicated” (L 116)

Results

   Change "familiar" to familial

Done (L 129)

CTV- did you retract for anatomical boundaries i.e did CTV extend outside the eye? the orbit? If group C eye did you use this 5 mm margin into vitreous ? or just around/along retina? 

Were apertures used?  Please comment on spot size unless you feel too much detail for readership

5 mm margins were used for the entire globe including the vitreous body. Apertures were only used in individual cases. Treatment was partly by uniform scanning with apertures, partly by pencil beam scanning with and without apertures. The group is very heterogeneous. (L 120-121)

Discussion

mention brain, muscle minimizations.  Most second malignancies in RT field are sarcomas or brain tumors

Second malignancies and brain tumors added to the discussion (L 249-250).

The authors could consider further discussion on the potential downside to delaying radiation, a clearly effective therapy, for years.  This leads to loss of vision, potential for metastatic spread.  This is based on fear of second malignancy from highly different radiation fields and the risk is much lower with small field proton RT.  Additional bony growth abnormalities are also so much lower.  When I see patients referred for proton RT now, they usually have so little vision and have gradually lost it trying treatments that may work for a period of time but then ultimately show progression and with this progression vision loss.  If the authors could add actual #'s to go with the sentence in the discussion "systemic chemotherapy in advanced tumor stages is often associated with a higher recurrence rate, especially if no local adjuvant therapy is used" this would be of interest to some readers. 

We appreciate the reviewer’s comment and we completely agree that proton therapy should not only be considered as a last resort when all other methods have been unsuccessful. This study, however, does not provide sufficient long-term data on recurrence rates after chemotherapy.  However, in the spirit of this reviewer's remarks, we have stated in our conclusions that radiation therapy should be considered as initial treatment option e.g. in small solitary tumors close to radiosensitive structures like fovea and optic disk and not only in massively pre-treated eyes with no other therapy option (L 375-380).

Reviewer 2 Report

This is a nicely illustrated and well written paper showing salvage rates for proton beam therapy with respect to heavily pre-treated eyes. The area is of interest to retinoblastoma specialists. The priorities for readers are  1)  what the follow-up is and 2) were there any metastases?

For 15 patients this should be relatively simple. However, in this manuscript it states minimum 12 months FU in the methods and 2 months in Table 2. Regarding metastases, one is mentioned, but in Table 2  two are mentioned. Readers would want more information about the 2nd patient with metastases. This gives a met rate of (2/15) 14% (not mentioned in abstract or paper).

The follow-up is too short for any discussion about SPT rate. The other PBT papers have looked at decades of follow-up.

P8 L272. Mouw et al’s abstract suggests there is a difference in orbital growth but the discussion does not. Authors would benefit reading this paper in its entirety as the findings are controversial.

Fig 3 Legend. Should add that this patient has recurrent corneal erosions after the transplant.

P10 L356 Conclusion should be based on the findings of this study not other publications.

Typo

P3 L125 ‘familial’ not ‘familiar’

Author Response

Thank you very much for your time and effort in revising our manuscript. We appreciate that and have implemented your suggestions for improvement wherever possible. 
Below you will find our revisions and responses to your suggestions:

The priorities for readers are  1)  what the follow-up is and 2) were there any metastases?

For 15 patients this should be relatively simple. However, in this manuscript it states minimum 12 months FU in the methods and 2 months in Table 2.

Unfortunately, some eyes had to be enucleated before the required follow-up had been completed. Nevertheless, we wanted to include them in our study population to allow for the most accurate assessment of PBT. The legend of Table 2 has been changed (L 206-208).

Regarding metastases, one is mentioned, but in Table 2  two are mentioned. Readers would want more information about the 2nd patient with metastases. This gives a met rate of (2/15) 14% (not mentioned in abstract or paper).

Thank you very much for this comment. A corresponding subitem in the results section has been added (L 235-243).

The follow-up is too short for any discussion about SPT rate. The other PBT papers have looked at decades of follow-up.

We fully agree with reviewer 2. We did not discuss the rate of SPTs in detail because of the short follow-up in our study. The primary concern was the feasibility of the method and construction of the smallest possible irradiation field.

P8 L272. Mouw et al’s abstract suggests there is a difference in orbital growth but the discussion does not. Authors would benefit reading this paper in its entirety as the findings are controversial.

The paper of Mouw et al. (2014) describes a subset of patients in whom the posterior chamber of the eye and the median orbital wall were included in the target volume in the case of larger tumours. In individual cases, growth restriction of the orbit had then occurred (Mouw et al. 2017). This point was added to the discussion in our manuscript (L 272).

Fig 3 Legend. Should add that this patient has recurrent corneal erosions after the transplant.

Done (L 226-227).

P10 L356 Conclusion should be based on the findings of this study not other publications.

Due to the short follow-up, we cannot present own data on growth inhibition of the bony orbit following PBRT, but we believe that this aspect of PBRT should be addressed in the summary within the discussion. The  discussion has been shortened according to the suggestion of the reviewer (L 379).

Typo

P3 L125 ‘familial’ not ‘familiar’

Corrected (L 125).

Reviewer 3 Report

Authors report their experience in proton beam therapy in retinoblastoma tumor. Due to radiation-related side-effects, which include secondary primary malignances, radiotherapy can be only used as a rescue therapy in retinoblastoma. Proton beam therapy method shows reduced secondary effects.

The paper is clear and well written.

Only fifteen cases occurring in thirteen patients are described in the present retrospective study, yet retinoblastoma is a rare tumor and proton beam therapy is a third line therapy. Despite the low number the conclusions of the study are sufficiently firm. Therefore, the data presented will be helpful for people involved in retinoblastoma therapy and even in other pediatric tumors.

I recommend publication in present form.

Author Response

Thank you very much for your time and effort in revising our manuscript. We appreciate that and have implemented your suggestions for improvement wherever possible. 
Below you will find our revisions and responses to your suggestions:

Authors report their experience in proton beam therapy in retinoblastoma tumor. Due to radiation-related side-effects, which include secondary primary malignances, radiotherapy can be only used as a rescue therapy in retinoblastoma. Proton beam therapy method shows reduced secondary effects.

The paper is clear and well written.

Only fifteen cases occurring in thirteen patients are described in the present retrospective study, yet retinoblastoma is a rare tumor and proton beam therapy is a third line therapy. Despite the low number the conclusions of the study are sufficiently firm. Therefore, the data presented will be helpful for people involved in retinoblastoma therapy and even in other pediatric tumors.

I recommend publication in present form.

Thank you very much for your time and the positive review of our manuscript.

Reviewer 4 Report

So fare, radiation is effective in RB, but has been marked due to secondary CNS tumor development afterwards. Proton beam radiation seems therefore an excellent option to treat RB eyes without burden the CNS with radiation dose.

Biewald et al. report retrospectively first results regarding proton beam radiation therapy to salvage RB eyes after previous max treatment. The data of this study are of highest clinical relevance for RB and are extremely valuable.

The IRB has been waived according to the retrospective character of the study, which is acceptable. The summary, the abstract, the M&M section als well the conclusion should inform about the retrospective character. Also the reasons for waiving  the IRB has to be explored in the M&M section.

Some reviewers might criticise the short term follow up of this study to detect secondary malgnicies and tissue malformation after growth. Therefore, to further inform, data from the PBT planing has to be included: a) radiation dose at a defined point in the CNS (e.g. Chiamsa opticum), b) at the orbital bones, c) at the optic nerve, d)at the lens, e) at the cornea. These data have to be also discussed with radiation dose from previous ERBT in RB. Has the study group seen any benefit regarding the PBT dose burden of the health tissue in and around the eye?

Standard Diviation values for age and mean follow has to be included. Also testing for Gaussian disturbution for age and mean follow might be tested.

Author Response

Thank you very much for your time and effort in revising our manuscript. We appreciate that and have implemented your suggestions for improvement wherever possible. 
Below you will find our revisions and responses to your suggestions:

The IRB has been waived according to the retrospective character of the study, which is acceptable. The summary, the abstract, the M&M section als well the conclusion should inform about the retrospective character. Also the reasons for waiving  the IRB has to be explored in the M&M section.

The retrospective study nature was added to the requested manuscript sections, as were the reasons for not obtaining an ethics vote (L 28, 39, 94-95).

Some reviewers might criticise the short term follow up of this study to detect secondary malgnicies and tissue malformation after growth. Therefore, to further inform, data from the PBT planing has to be included: a) radiation dose at a defined point in the CNS (e.g. Chiamsa opticum), b) at the orbital bones, c) at the optic nerve, d)at the lens, e) at the cornea. These data have to be also discussed with radiation dose from previous ERBT in RB. Has the study group seen any benefit regarding the PBT dose burden of the health tissue in and around the eye?

We thank reviewer 4 for this comment, which emphasizes the undoubtedly important comparison between conventional radiation and PBRT. Comparative planning with conventional radiotherapy, however, is scheduled as a separate study as it would go beyond the scope of this study.

Standard Diviation values for age and mean follow has to be included. Also testing for Gaussian disturbution for age and mean follow might be tested.

Added without Gaussion distribution (L 127, 130).

Round 2

Reviewer 2 Report

Please add the metastatic rate in the abstract. Then this paper can be accepted

Author Response

Thanks again for the helpful comments and remarks.

The metastatis rate was added to the abstract.
